# Drivers’ Comprehensive Emotion Recognition Based on HAM

**DOI:** 10.3390/s23198293

**Published:** 2023-10-07

**Authors:** Dongmei Zhou, Yongjian Cheng, Luhan Wen, Hao Luo, Ying Liu

**Affiliations:** 1School of Mechanical and Electrical Engineering, Chengdu University of Technology, Chengdu 610059, China; zhoudm@cdut.edu.cn (D.Z.); wenluhan@stu.cdut.edu.cn (L.W.); luoh@stu.cdut.edu.cn (H.L.); 2China Unicom Digital Technology Co., Ltd. Hubei Branch, Wuhan 430015, China; liuying311@chinaunicom.cn

**Keywords:** driver emotion, multimodal fusion, CNN, Bi-LSTM, HAM

## Abstract

Negative emotions of drivers may lead to some dangerous driving behaviors, which in turn lead to serious traffic accidents. However, most of the current studies on driver emotions use a single modality, such as EEG, eye trackers, and driving data. In complex situations, a single modality may not be able to fully consider a driver’s complete emotional characteristics and provides poor robustness. In recent years, some studies have used multimodal thinking to monitor single emotions such as driver fatigue and anger, but in actual driving environments, negative emotions such as sadness, anger, fear, and fatigue all have a significant impact on driving safety. However, there are very few research cases using multimodal data to accurately predict drivers’ comprehensive emotions. Therefore, based on the multi-modal idea, this paper aims to improve drivers’ comprehensive emotion recognition. By combining the three modalities of a driver’s voice, facial image, and video sequence, the six classification tasks of drivers’ emotions are performed as follows: sadness, anger, fear, fatigue, happiness, and emotional neutrality. In order to accurately identify drivers’ negative emotions to improve driving safety, this paper proposes a multi-modal fusion framework based on the CNN + Bi-LSTM + HAM to identify driver emotions. The framework fuses feature vectors of driver audio, facial expressions, and video sequences for comprehensive driver emotion recognition. Experiments have proved the effectiveness of the multi-modal data proposed in this paper for driver emotion recognition, and its recognition accuracy has reached 85.52%. At the same time, the validity of this method is verified by comparing experiments and evaluation indicators such as accuracy and F1 score.

## 1. Introduction

For motor vehicle driving, negative emotions are one of the important factors leading to traffic accidents. Several studies have shown that emotions can affect a driver’s decision-making ability, alertness, and driving behavior. Road rage and drowsy driving are simple examples of how emotions can affect driving behavior, but negative emotions such as sadness and fear, as well as being overexcited, are also important factors in driver behavior. According to a survey, about 1.25 million people die in traffic accidents every year, and an average of 3400 people die on the road every day [1]. The government can take some mandatory measures to restrain dangerous driving behaviors such as drunk driving, but it cannot restrain drivers’ emotional problems. Studies of vehicle crashes have found that when driver-related warning systems can alert drivers of potential hazards, the likelihood of injury-related crashes decreases. Therefore, it is of great value to improve driving safety to develop an automatic recognition of drivers’ emotions with a high feasibility and accuracy and a corresponding classification early warning system, and to adopt corresponding adjustment strategies to alleviate drivers’ negative emotions and avoid traffic accidents.

In recent years, in order to reduce traffic accidents and improve road safety, the research on drivers’ emotions has received more and more attention [2,3,4]. Studies have shown that the signals related to driver emotion recognition can be roughly divided into physiological signals and non-physiological signals. Physiological signals mainly include electroencephalography (EEG) [5,6], electrocardiography (ECG) [7], and electromyography (EMG) [8]. Audio and video signals, facial expressions, blink rate, body posture, and driving vehicle data are non-physiological signals. Based on emotion-related signals, researchers have conducted extensive research on emotion recognition [9,10,11,12], using single-modal or multi-modal data to identify certain negative emotions of drivers, and proposed a good research method in enhancing emotion recognition, effectively improving driving safety [13,14,15].

In the study of driver emotions, many studies have used the powerful nonlinear modeling ability, automatic feature extraction ability, and large-scale data processing ability of neural network as effective tools for emotion recognition. The convolutional neural network (CNN) is widely used in image processing and natural language processing [16,17], and it also performs well in sequence data processing. For emotional state recognition tasks, input signals may come from various sensors, such as cameras, microphones, and heart rate meters; hence, the input data has multi-dimensional feature representations, and the CNN has natural advantages in processing multi-dimensional features. The CNN can effectively extract local features and retain global information, showing good performance in human expression and voice recognition. Secondly, the bidirectional long short-term memory network (Bi-LSTM) [18,19,20] is a common sequence modeling method, which can capture the long-term dependencies of the input sequence, thereby improving the effect of sequence modeling and processing. In driver emotional state recognition, the Bi-LSTM can learn potential temporal patterns and semantic information, and capture temporal information to further improve the accuracy of emotional state classification. Finally, the introduction of the hybrid attention module (HAM) can combine different levels and different types of features to improve the robustness and generalization ability of the model [21,22]. This mechanism is suitable for multimodal data and complex tasks. In common driver facial emotion recognition tasks, it is usually necessary to model facial expression features and predict a driver’s emotional state. However, different features in multimodal data have different importance in different emotion recognition; thus, the attention mechanism needs to be adjusted dynamically to adaptively select key facial features [23,24]. Therefore, the HAM technology is proposed for drivers’ comprehensive emotion recognition to solve this problem. It can consider different levels and different types of features at the same time to improve the performance and robustness of drivers’ comprehensive emotion recognition.

The recognition and analysis of drivers’ emotions is very important in the real driving environment. However, most of the previous studies used a single modality to monitor negative emotions, or monitored a certain negative emotion through multimodal data such as physiological signals; however, these methods have limitations [8,25]. In addition, physiological signal acquisition equipment is expensive and inconvenient to wear, which further limits its application. In order to solve these problems and supplement the deficiencies of the current research, this paper proposes a driver emotion recognition classification model based on the CNN + Bi-LSTM + HAM, aiming to pursue a more comprehensive and accurate real-time recognition of a driver’s emotional state.

The main contributions of this paper are as follows:

In this paper, drivers’ comprehensive emotion recognition is based on the idea of multi-modality. Combining three modalities of driver voice, facial image, and video sequence using three datasets of the RAVDESS [26], CK+ [27], and NTHU [28] for feature fusion, six classification tasks of driver emotions are identified as follows: sadness, anger, fear, fatigue, happiness, and emotional neutrality.

In view of the complexity of drivers’ multimodal data and emotion recognition tasks, and in order to further improve the classification accuracy of emotional state recognition, this paper introduces the HAM on the basis of the CNN + Bi-LSTM network framework. The mechanism can consider different levels and types of features at the same time, and it can dynamically adjust the attention mechanism according to different situations, and adaptively select key facial features. Therefore, this paper proposes a driver emotion recognition model based on the CNN + Bi-LSTM + HAM.

The validity of the model is verified with experiments. Fusion is performed at the feature layer to generate a multi-modal fusion vector, which is then transmitted to the CNN + Bi-LSTM + HAM network to successfully identify the specific emotions of the driver.

## 2. Related Work

In order to describe human emotions, psychological researchers have proposed the discrete emotion theory and the dimensional emotion theory to classify emotions. Currently, the most famous discrete emotion model is the basic emotion model proposed by Ekman [29]. Other emotions are considered as combinations of these basic emotions, and this paper uses a discrete emotion model to describe emotions quantitatively. Driver emotion recognition is usually achieved by analyzing a driver’s emotional representation. Representations of human emotions can include facial expressions, language, body posture, and physiological changes [30,31]. In this research on driver emotion recognition, a variety of different measurement methods are used [32]. These include methods such as facial expression analysis, speech analysis, observation of driving behavior, and measurement of physiological signals [33,34]. These methods can all provide clues about the driver’s emotional state. Facial expressions play an important role in emotion recognition. According to different research studies, facial expression is one of the crucial factors in human communication. Verbal components convey only one-third of human communication, while non-linguistic components convey two-thirds of human communication. Among the many non-verbal components, facial expression has become one of the main information channels in interpersonal communication by expressing emotional meaning. Therefore, it is of great significance to use facial expression analysis as a means of drivers’ emotion recognition. By recognizing and understanding the facial expressions of drivers, it is possible to better understand their emotional state and take corresponding measures to improve driving safety and comfort.

Emotional state has a significant impact on drivers’ judgment and behavior during driving. Anger, sadness, fear, and other emotional states that are not conducive to driving can easily lead to wrong judgments by drivers and thus endangering road traffic safety. Therefore, incorporating driver emotion recognition into intelligent assisted driving systems has broad application prospects. Katsis et al. conducted a study using various physiological indicators such as facial electromyography, electrocardiogram, respiration, and electrodermal activity to assess the emotional activity of drivers under racing simulation conditions [35]. They used support vector machines and an adaptive neuro-fuzzy inference system to classify the data and proposed a method for assessing the emotions of racecar drivers. They also designed a wearable system, including a multi-sensor wearable module, a centralized computing module, and a system interface. Orit Taubman Ben-Ari et al. conducted multigroup trials to develop a multidimensional driving style scale for assessing driving style, covering four broad domains [36]. They analyzed the relationship between different driving styles and personality indicators such as driver status and found associations between measures of trait anxiety and neuroticism. The results of their study showed that under the influence of positive emotions, the driving behavior of male test drivers may be bolder. These studies show that emotion recognition has an important application value in intelligent assisted driving systems. By monitoring a driver’s emotional state, the system can promptly remind the driver to pay attention to the emotional impact, and adjust the auxiliary measures according to the emotional changes to improve driving safety and effectiveness. However, further research and development are needed to apply emotion recognition technology to real driving scenarios and to address related technical, privacy, and ethical issues.

In recent years, driver emotion analysis using single data and multimodal data is also a popular research direction in recent years. Most researchers focus on single modality data, such as physiological data for emotion recognition. Emotion recognition based on EEG signals is currently one of the most popular research areas. Nakisa reviews State-of-the-Art EEG-based features to classify different emotions [37]. However, in complex situations, a single modality may provide poor robustness. Therefore, some researchers use different signaling patterns to recognize human emotions. For example, Soleymani uses EEG and pupil data for emotion recognition [38]. Ghaleb has achieved better results than ever before by fusing video and audio data for emotion recognition through a multimodal fusion framework [39]. Moreover, Tavakoli proposed a framework called HARMONY and explored multimodal longitudinal driving research in real driving scenarios [40]. Nakisa’s research shows that the emotion recognition performance of multimodal data fusion is higher than that of a single modality. Although great progress has been made in emotion recognition, the data used in many studies are invasive, expensive, and inconvenient to wear in real-world driving situations. Furthermore, noninvasive multimodal data with characteristics of driving situations has far fewer applications in identifying driver emotions than physiological data and needs to be further explored.

Some progress has been made in previous research work, but there are still some challenges and limitations. First of all, emotion recognition is a complex task that involves the comprehensive consideration of multiple factors, such as driving environment, individual differences, and the interaction between different emotions. For example, a driver may display nervousness and anxiety in a high-pressure driving environment, but joy and relaxation in a relaxed environment. The complexity of these factors makes emotion recognition difficult and requires more in-depth research to understand and address. Second, most current research focuses on emotion recognition based on a single or a few physiological signals, such as heart rate, galvanic skin response, etc. However, recognition only relying on a single signal may not be able to capture a driver’s emotional state fully and accurately. Therefore, fusion analysis of multi-source information is a challenging task. In addition, introducing an attention mechanism in emotion recognition is also a potential research direction. Current research mainly focuses on introducing attention mechanisms into deep learning-based models to improve the model’s attention to key information. However, the design and optimization of the attention mechanism still needs to be further explored, especially in the application scenarios in the field of emotion recognition. For example, how to design an effective attention mechanism based on the characteristics and representations of different emotions, and how to solve the problems that the attention mechanism may encounter in practical applications, etc. In this paper, we propose a CNN + Bi-LSTM + HAM network model, which aims to overcome the above problems and effectively recognize a driver’s comprehensive emotional state. The model combines the CNN and Bi-LSTM to capture the spatial and temporal correlations in the input data, and utilizes the HAM to improve the model’s attention to key information. We believe that the application of this comprehensive model will be able to identify drivers’ emotions more accurately, and provide new ideas and methods for further emotion recognition research.

## 3. Materials and Methods

In order to realize comprehensive driver emotion recognition and improve driver safety, this paper detects drivers’ negative emotions by combining drivers’ voice, static facial images, and video sequences to improve drivers’ safety. In this paper, three dataset modalities, the RAVDESS, CK+ and NTHU, are selected. First, for the audio modality of the RAVDESS dataset, the hybrid mel-frequency cepstrum coefficients (MFCC) method is used to extract the audio feature vector. The hybrid MFCC is an improved audio feature extraction method for extracting feature vectors of audio signals [41,42]. Compared with the traditional MFCC method [43,44], the hybrid MFCC can extract a variety of features with different frequencies, thereby improving the accuracy and stability of the task. Second, for the static facial image state of the CK+ dataset, the facial feature vector of the static image is extracted using the histogram of oriented gradients (HOG) method [45,46]. The HOG is a commonly used image feature extraction method, which can capture the edge and texture information of objects in the image. Finally, for the video sequence modality of the NTHU dataset, the circular local binary patterns algorithm is used to extract the video sequence feature vector [47]. Compared with the traditional local binary patterns (LBP) algorithm [48,49], the circular LBP algorithm is more flexible and adaptable in describing the statistical characteristics of local texture in images or videos. It can better capture texture information of different scales and directions, and can also adjust parameters to meet the needs of specific tasks.

In the feature layer, the feature vectors extracted from different modalities are spliced to form a fusion feature vector that is passed into the CNN + Bi-LSTM + HAM network for driving comprehensive emotion recognition task. The CNN is mainly used to extract features. The convolutional layer can locally sense the input data through a sliding window, thereby capturing the spatial features in images or other state data. Through the stacking of multiple convolutional layers and pooling, the CNN can gradually extract higher-level features. The Bi-LSTM is used to model sequence data, which can effectively deal with the time series dependence of sequence data. Through the LSTM units in both forward and reverse directions, the Bi-LSTM can consider both past and future information mentioned above, leading to a better understanding of the semantics and structure in sequence data. Finally, the HAM is used to strengthen the attention weight adjustment of key features, which can automatically learn and adjust the weights of different features according to the importance of input data, to enable the network to pay more attention to the information useful to the current task. In order to avoid the overfitting problem, we added a dropout layer [50,51] between the fully connected layer and the Bi-LSTM layer to reduce the complexity of the neural network, which can improve the generalization ability of the model and increase the accuracy of the model. In general, the model proposed in this paper combines the features of different modalities, and uses the CNN, Bi-LSTM, and HAM for feature extraction, sequence modeling, and attention weight adjustment, to effectively realize the task of comprehensive driving emotion recognition; the overall implementation framework of driver comprehensive emotion recognition is shown in Figure 1.

### 3.1. Model Framework for Comprehensive Emotion Recognition of Drivers

#### 3.1.1. CNN

The CNN is widely used in image processing and natural language processing, and has also shown good performance in sequence data processing. Due to the multidimensional feature representation of input data, the CNN naturally excels in handling multidimensional features, enabling effective extraction of local features while preserving global information. It demonstrates good performance in recognizing human expressions and sounds, among other aspects. As shown in Figure 2, the CNN, as a type of multilayer perceptron, consists of convolutional layers, pooling layers, and fully connected layers. The pooling layer effectively reduces the number of weights, making the network easier to optimize and reducing the risk of over-fitting.

The CNN performs a convolution operation on the input signal through a convolutional layer to achieve spatial translation invariance. This enables the CNN to capture local patterns when processing image and spatial data. By sharing weights and using pooling layers, the CNN is also able to have a certain degree of invariance to transformations such as translation, rotation, and scaling. The pooling layer and fully connected layer of the CNN can aggregate the adjacent features and global features, respectively, to conduct a comprehensive analysis of the speech signal. In the CNN model, feature extraction is performed automatically by the convolutional layer without manual intervention, which makes it easier for the CNN to extract feature information suitable for machine recognition. Compared with traditional manual feature extraction methods, the CNN can obtain better feature generalization performance. The CNN model performs well in image processing, can obtain good feature information from images, and has achieved remarkable results in tasks such as emotion classification and facial expression recognition. The method based on the CNN model and its optimization and improvement has achieved remarkable accuracy in facial expression recognition of static images.

#### 3.1.2. Bi-LSTM

Long short-term memory network (LSTM) [52,53] is mainly used in scenarios with time series characteristics, such as feature extraction for speech. However, the LSTM has a problem when modeling input with time series features, which is that it cannot encode information from a backward to a forward direction. In the emotion six classification task discussed in this article, if more fine-grained classification is required, we need to pay attention to the relationship between words. Specifically, we need to focus on vocabulary such as emotional words, profane words, and negative words. In the process of emotion recognition, the use of the bidirectional long-short-term memory network (Bi-LSTM) can solve the problem of the LSTM, which is that it cannot access contextual information at the same time. The Bi-LSTM allows forward and backward propagation of information, processing both past and future contextual information at each time step. This enables the Bi-LSTM to better capture the word-to-word dependencies in time series data, improving the accuracy of sentiment classification. By using the Bi-LSTM, the input with time series features can be effectively processed, and the association between words can be better focused on, which helps to improve the performance of emotion recognition.

The Bi-LSTM is a commonly used sequence modeling method, which can capture the long-term dependencies of the input sequence, thereby improving the effect of sequence modeling and processing. In driver emotional state recognition, the Bi-LSTM can learn potential temporal patterns and semantic information, and can also capture temporal information, thereby further improving the classification accuracy of emotional states. The Bi-LSTM is composed of two LSTMs, which are carried out from the forward and backward directions during the training process. Finally, the obtained hidden layers are concatenated to obtain the output layer. A structure diagram of the Bi-LSTM network is shown in Figure 3.

#### 3.1.3. HAM

The CNN + Bi-LSTM model is a deep learning model commonly used in emotion recognition tasks. It combines the CNN and Bi-LSTM to fully capture the contextual information in the text and model local features. The Bi-LSTM is able to process longer text sequences, maintaining the ability to model long-distance dependencies. By combining these two models, the spatial and temporal characteristics of drivers’ emotional state can be considered simultaneously, improving the model’s ability to understand their emotional state. However, the CNN + Bi-LSTM model usually requires a lot of computing resources and training time. The training process of the model can be very time-consuming, especially for large-scale driver emotion datasets,. In addition, there are a large number of hyper-parameters in the CNN + Bi-LSTM model that need to be tuned, such as the size of the convolution kernel, the pooling operation, the number of LSTM hidden layer units, etc. The selection of these hyper-parameters is critical to model performance, but the tuning process is cumbersome. In the driver emotional state recognition task, we need to accurately capture and distinguish different emotional states from multiple data sources. Although the CNN and Bi-LSTM are excellent in image and sequence data processing, only using the CNN + Bi-LSTM may not have the capacity to fully capture the characteristics of all emotional states. As emotional states involve multiple perception channels, such as voice, facial expression, etc., these perception channels may show different patterns and differences in different emotional states. Therefore, introducing the HAM module into the driver emotion classification model can provide the model with additional image feature extraction and classification capabilities.

The HAM is a very lightweight yet effective attention module. It receives intermediate feature maps as input and generates channel attention maps and channel refinement features via the channel attention module (CAM) [21]. Then, according to the channel attention map, the spatial attention module (SAM) divides the channel refinement features into two groups and generates a pair of spatial attention descriptors. Finally, with spatial attention descriptors, the SAM generates final refined features that can adaptively emphasize important regions. The HAM is a simple, general-purpose module that can be seamlessly embedded into various mainstream deep CNN architectures, and can be trained end-to-end with an end-to-end base CNN. Since both channel attention and spatial attention are important to the final goal, the HAM module applies channel attention and spatial attention sequentially, which can help the network to effectively learn which features to emphasize or suppress. Figure 4 shows the structure of the HAM, which consists of two sub-modules, the CAM and SAM. The CAM can generate 1D attention maps, while the SAM can generate a pair of 2D attention maps. The intermediate features can be further optimized through the refinement of the HAM in terms of channel and space, which allows it to be embedded in any State-of-the-Art deep CNN. Figure 4 shows the HAM network structure.

The channel attention module (CAM) [22,54,55]: The CAM plays an important role in deep learning networks. It selects the most important channel features through the learned weights to enable the model to better understand the relationship between different channels. The goal of the CAM is to focus attention on more important content in the input image, thereby achieving the aggregation of spatial dimension information. To generate channel attention maps, we need to exploit the relationship between channels in the feature map. The channels of each feature map can be viewed as a feature detector, while channel attention focuses on meaningful content in a given input image. To efficiently compute channel attention, the spatial dimension of the input feature map is usually compressed. In terms of aggregation of spatial information, average pooling is a commonly used method [56]. Max pooling is also able to gather important cues about unique object features and thus infer finer channel attention. Therefore, at different stages of image feature extraction, average pooling and max pooling play different roles in balancing spatial information and unique feature aggregation. In order to distinguish the features obtained with average pooling and maximum pooling, we can use an adaptive mechanism. In this way, average pooled and max pooled features should not be weighted as the same feature. A common approach is to use 1D convolutions to avoid channel dimensionality reduction and capture cross-channel interactions. By flowing rich features into a fast one-dimensional convolution, and then activating the Sigmoid function, the channel attention map is finally generated. This preserves important channel features and further enhances its representational power. Figure 5 shows a CAM network structure diagram.

The spatial attention module (SAM) [54,57]: The SAM can be used to focus on the attention mechanism of important regions in a driver’s facial image. The SAM performs attention weighting on spatial dimensions to help the model better focus on important regions. In the SAM, channel attention and spatial attention are two sub-modules, which focus on different attention information, respectively. The CAM generates a channel attention map showing the importance of different channels. This graph clearly tells us the relative importance of each channel in the feature representation. The channel attention focuses on the “what” more important information. The SAM exploits the inter-spatial relationship of features to generate a spatial attention map. In contrast to the channel attention, spatial attention focuses on “where” more important information is located. It complements the role of channel attention. To compute spatial attention, average pooling and max pooling operations are first applied along the channel axis and then concatenated to generate an efficient feature descriptor. Next, using the channel separation technique, the features are divided into two groups according to the channel attention map. Then, the two sets of features are average-pooled and max-pooled along the channel axis, and the output is passed to a shared convolutional layer. With the SAM, the model is able to better focus on important regions in a driver’s facial image, improving the performance of driver monitoring and analysis. Figure 6 shows a SAM network structure diagram.

Various types of data are involved in driver emotion recognition, such as images, voices, etc. Different emotional states may have differences in time and space. The HAM can apply different attention mechanisms to different types of data, thereby improving feature extraction and relational modeling that handle these heterogeneous data well. The HAM has the characteristic of feature interaction, which can promote the interaction and information transfer between features by modeling the correlation between multiple features. In driver emotion recognition, different types of features, such as voice, facial expression, etc., may have different importance. Through the HAM, the weight of different features can be flexibly adjusted to improve the model’s perception of key features. The HAM has high flexibility and adaptability, and it can dynamically adjust the weights of different attention mechanisms according to specific situations to enable the model to better adapt to the expressions and important features of different emotional states. Overall, the HAM has stronger feature integration ability, multi-scale attention ability, flexibility and adaptability in the driver emotion recognition model, which can improve the performance and robustness of the model.

### 3.2. CNN + Bi-LSTM + HAM Network Model to Recognize Driver’s Comprehensive Emotions

In this paper, we utilize three dataset modalities of the RAVDESS, CK+, and NTHU for the comprehensive driver emotion recognition task. First, we use the hybrid MFCC method to extract audio feature vectors for the audio modality in the RAVDESS dataset. For the facial image modality of the CK+ dataset, we use the HOG algorithm to extract feature vectors of static facial video images. Finally, for the video sequence modality of the NTHU dataset, we employ the circular LBP algorithm to extract feature vectors. In order to comprehensively utilize the information of different modalities, we concatenate the feature vectors extracted from different modalities at the feature layer to form a fusion feature vector. Doing so allows for a more complete picture of the driver’s emotional state and improves the accuracy of emotion recognition. In terms of model design, we adopted the CNN and Bi-LSTM to process the fused feature vectors. The CNN is widely used in image processing and natural language processing, and has natural advantages in processing multi-dimensional features. It can effectively extract local features and preserve global information, which is very suitable for driver emotion recognition tasks. The Bi-LSTM, on the other hand, can capture the long-term dependencies of the input sequence and further improve the classification accuracy of emotional states. To enhance the robustness and generalization ability of the model, we introduce the HAM. The HAM can fuse different levels and different types of features to better utilize the information of multimodal data. This can further improve the classification accuracy of the driver’s emotional state recognition task. Finally, we process the final features with a fully connected layer to obtain a prediction of the driver’s emotional state. In order to transform the prediction result into a probability distribution, we input it into the Softmax activation function to obtain the final driver emotion classification result [58,59]. This model comprehensively utilizes the information of multi-modal data and can identify a driver’s emotional state more accurately. Figure 7 shows the structure of the CNN + Bi-LSTM + HAM network model.

During the training process, in order to solve the problems of high computational resources and time consumption and overfitting, we improved the model. A dropout layer is introduced between the fully connected layer and the Bi-LSTM layer to reduce the complexity of the neural network and improve the generalization ability and accuracy of the model. Dropout is a widely used regularization technique that can effectively reduce the risk of overfitting. The main function of the dropout layer is to randomly set the output of some neurons to zero in each training batch to enable the model to learn more different features and reduce the interdependence between neurons, thereby improving the robustness of the model, adherence, and generalization ability. Its working principle is that during the neural network training phase, dropout will temporarily discard certain nodes in the hidden layer of the network with a certain probability. These discarded nodes can be considered as not being a part of the network, allowing other nodes to focus on training and only update the parameters of these “worker” nodes. Therefore, each training is equivalent to training a different network structure. Most experiments show that dropout has the ability to prevent model overfitting. By introducing the dropout layer, the complexity of the neural network can be reduced, the generalization ability of the model can be improved, and the accuracy of the model can be increased. Figure 8 shows a working diagram of dropout.

Dropout has the ability to prevent the model from overfitting. The essential reason is that the network with random behavior is trained through dropout, and multiple random decisions are averaged to enable each hidden layer to learn to perform well under different conditions, forcing it to learn more salient features.

Activation function: Neural network models need to learn and understand complex nonlinear functions through nonlinear mapping. Activation functions play a crucial role in this process. It introduces nonlinear characteristics, which makes the neural network have more powerful expressive ability, and can better adapt and fit complex data. If no activation function is used, the output of the neural network will simply be a simple linear function, similar to a linear regression model. This kind of model has limited ability and can only solve simple problems, but cannot cope with complex data and tasks. By using activation functions, neural networks can introduce nonlinear transformations, enabling them to capture complex patterns and nonlinear relationships in data. The nonlinear nature of the activation function enables the neural network to learn and represent more complex functions, improving the expressive and learning capabilities of the model. Therefore, the importance of activation functions for neural networks cannot be ignored. Choosing an appropriate activation function depends on the specific task and the nature of the dataset. Common activation functions include the Sigmoid function [60], Tanh function [61], and ReLU function [62,63]. According to the actual situation, you can also choose other activation functions or their variants to further improve the performance of the model. In order to avoid problems such as gradient dispersion caused by the activation function, this paper chooses the ReLU function. Compared with other activation functions, its structure is simpler and more effective. Its basic formula is shown in Equation (1) as follows:(1)$$ReLU(x)=\left\{\begin{array}{c} x\;\;,\;\;\;\mathrm{if}\;\;\;x\geq 0\\ 0\;\;,\;\;\;if\;\;\;x\leq 0 \end{array}\right.$$

In Formula (1), we can find that the gradient of the ReLU function is 1 when it is positive and 0 when it is negative, which makes it perform well in solving the problem of gradient disappearance. Since the derivative of the ReLU function is constant at 1 in the positive interval, it is possible to avoid gradient decrement or gradient explosion during backpropagation. Therefore, the ReLU function is widely used in deep neural network models.

The loss function: The loss function, also known as the cost function, is a function that is usually used for parameter estimation and to evaluate the difference between the estimated value and the real value of the data. Through the loss function, the training problem of the network can be converted into a function optimization. However, compared with using the mean squared error (MSE) [64], when the model is first trained, it is often found that when the output probability of the model is close to 0 or 1, the partial derivative value will become very small, resulting in a slow update of parameters and failure of the model to converge. Therefore, this article chooses the cross-entropy function [65,66]. The cross-entropy loss function is shown in Equation (2) as follows:(2)$$L=\frac{1}{N}\sum_{i}^{}-[y_{i}log(p_{i})+(1-y_{i})log(1-p_{i})]$$

In Formula (2), $y_{i}$ represents the label of sample i, the positive class is 1, and the negative class is 0; $p_{i}$ represents that sample i is predicted to have positive class probability.

## 4. Experiment and Results

### 4.1. Dataset

#### 4.1.1. The Ryerson Audio-Visual Database of Emotional Speech and Song: RAVDESS [26]

The Ryerson Speech Dataset, also known as the Ryerson Emotional Language and Song Audiovisual Dataset, was released in 2013. This is a large-scale audio emotion dataset developed by the Intelligence Lab team. The dataset was collected from 24 professional actors with pure North American pronunciation, including 12 male professional actors and 12 female professional actors. Several voice recordings and song recordings were recorded for each actor. The research in this paper only uses the speech audio samples in this dataset. A total of 24 professional actors participated in the speech audio collection. A total of 60 samples were collected from each person, with a total of 1440 samples. The composition of emotion tags is detailed in Table 1.

The seven emotions in Table 1: happiness, sadness, anger, fear, disgust, surprise, and calm all have two emotion intensities: normal and strong. The number of effective samples for model training and verification of the above seven emotions is 1344. There is also a neutral emotion, which is not distinguished by intensity. It should be noted that since we are focusing on the drivers’ six-category emotion analysis, we only use samples of five emotions: sadness, anger, fear, fatigue, happiness, and neutral, and form six-category multi-mode with other datasets. The emotion dataset is used to train the model in this paper.

#### 4.1.2. Video Modality Dataset: The Extended Cohn–Kanade: CK+ [27]

The extended Cohn–Kanade (CK+) dataset was released in 2010. It is a video sequence dataset developed by Cohen and Kanade et al. and is applied in the field of emotion recognition. This dataset has been optimized and refined to address three limitations of the Cohn–Kanade dataset published by the same team in 2000 and is used as the video modality in the multimodal dataset in this study. The data were collected from 123 subjects with a total of 593 video sequence samples containing human facial expressions, including 7 emotions such as happiness, sadness, anger, fear, disgust, surprise, and neutral.

The expression of facial emotions is very complicated. The expression of a specific emotion must have a process from brewing to eruption and then to fading. No emotion of any kind is displayed without warning. Emotional expression is influenced by many factors. In this paper, 593 video sequences in the CK+ dataset were retrieved using an artificial FACS encoder, and the occurrences of these 7 emotions were counted by the peak frames of the emotional outbursts in the video sequences. The distribution of 7 emotions in 593 video sequences is shown in Table 2. It should be noted that since we are focusing on drivers’ six-category emotion analysis, we only use the following samples of five emotions: sadness, anger, fear, fatigue, happiness, and neutral. We also form a six-category multi-mode with other datasets. The emotion dataset is used to train the model in this paper.

#### 4.1.3. National Tsing Hua University Dataset: NTHU Dataset [28]

The entire dataset (including training, evaluation, and testing datasets) contains 36 subjects of different ethnicities recorded with and without glasses/sunglasses in various simulated driving scenarios, including normal driving, yawning, and slow blinking, falling asleep, laughing, and more, in both day and night lighting conditions. Subjects were recorded while seated in a chair and played a simple driving game that simulated driving a wheel and pedals; at the same time, the experimenter instructed them to make a series of facial expressions. The total time for the entire dataset is about nine and a half hours. The training dataset contains 18 subjects in 5 different scenes (BareFace, Glasses, Night_BareFace, Night_Glasses, Sunglasses). Each subject’s sequence, which includes slow blink rates while yawning and nodding, is recorded for approximately 1 min. Sequences corresponding to the two most important scenes, the combination of sleepiness-related symptoms (yawning, nodding, slow blinking rate) and the combination of non-drowsiness-related actions (talking, laughing, looking to the sides), are each recorded for approximately 1.5 min. The test dataset consists of 360 videos for the completion of the dataset.

Furthermore, to simulate a more realistic driving situation, 18 subjects are randomly selected from the proposed dataset, but the group maintain a balance of various genders and skin colors. Their sequences were edited and combined into 2–10 min hybrid videos of each subject in five scenarios containing a varying number of transitions from non-sleepy to sleepy states or from sleepy to non-sleepy states of various situations. In total, 90 hybrid videos were added to the dataset for evaluation. It should be noted that since we are focusing on the six-category emotion analysis of drivers, we choose the NTHU dataset as the fatigue training dataset and as the fatigue emotion. The dataset is supplemented with six categories of emotions of drivers. At the same time, in order to perform dataset fusion with the RAVDESS dataset and CK+ dataset, this paper selects 144 video training datasets and 36 evaluation test datasets from two subsets of BareFace and Glasses. Figure 9 shows the simulated driving conditions of different people in different scenarios.

#### 4.1.4. Fer2013 Dataset [67]

The Fer2013 facial expression dataset is relatively large compared to other expression datasets. The pictures are closer to the daily life environment and are interfered with by various occlusions and other factors. It is a good choice for expression recognition research. The dataset consists of 35,886 facial expression pictures. The dataset is divided into three categories. There are 28,708 pictures in Training, and 3589 pictures each in PublicTest and PrivateTest. Each picture is composed of a grayscale image with a fixed size of 48 × 48 and a total of seven kinds of expressions. In order to fully discuss the results of the network proposed in this article on emotion recognition of a single modal dataset, and also to make up for the limitations of the imbalanced distribution of the multi-modal dataset used in this article, combined with the six categories of driver comprehensive emotions determined in this article, we selected the five emotions of sadness, anger, fear, happiness, and neutral from the public test set of the FER2013 dataset for experimental verification.

### 4.2. Dataset Feature Extraction

#### 4.2.1. Hybrid MFCC Extracts Features of RAVDESS Dataset

To realize the drivers’ comprehensive emotion recognition task, we adopt the emotion-rich RAVDESS dataset as the speech modality. In order to realize the task of emotion recognition and classification, it is first necessary to preprocess the input speech signal. Preprocessing methods such as pre-emphasis and frame windowing reduce the negative impact caused by the aliasing phenomenon of the voice signal and high-frequency harmonic distortion caused by the human vocal organ itself and the device that receives the voice signal, making it even and smooth.

In order to effectively extract speech features that distinguish emotions, represent emotions, and improve system robustness, some studies use the MFCC for feature selection. When selecting features, it is hoped that the features can represent the speaker’s emotion and have strong anti-noise ability. The MFCC can represent the auditory characteristics of human beings. First, the linear spectrum is mapped to the Mel nonlinear spectrum that can reflect human hearing, and then it is converted to the cepstrum. Cepstrum low-frequency information can reflect the envelope, while retaining some dimensions can achieve data compression.

Davies and Mermelstcin proposed $f_{Mel}$ using a logarithmic expression to simulate the nonlinear perception of the human ear to sounds of different frequencies [68]. Formula (3) is the conversion relationship between the two.
(3)$$f_{Mel}=1125\times \ln\left(1+\frac{f}{700}\right)$$

The MFCC is based on the cepstrum parameters derived from Mel frequencies. The Mel filter bank is a plurality of band-pass triangular filters set within the specified spectrum region, and the response function $H_{m}(K)$ is shown in Formula (4), where 0≤m≤M, and M is the number of filter Mel filters, and its center frequency is f(m).
(4)$$H_{m}(K)=\left\{\begin{array}{c} 0\;\;\;\;\;\;,\;\;\;\;\;\;k<f(m-1)\\ \frac{k-f(m-1)}{f(m)-f(m-1)},\;f(m-1)<k<f(m)\\ \frac{f(m+1)-k}{f(m+1)-f(m)},\;f(m)<k<f(m+1)\\ 0\;\;\;\;\;\;,\;\;\;\;\;k>f(m=1) \end{array}\right.$$

Equation (5) is the definition of f(m) as follows:(5)$$f(m)=\left(\frac{N}{F_{s}}\right)B^{-1}(B(f_{l})+m\frac{B{(f}_{h})-B{(f}_{l})}{M+1})$$

In Formula (5), $f_{h},\;f_{l}$ are the frequency domain ranges where the filter is located, representing the lowest and highest frequencies, respectively; N is the period range of discrete Fourier transform (DFT); $F_{s}$ is the sampling rate; and $B^{-1}$ is the inverse function of B.

The advantage of the MFCC is that it is similar to the way the human ear acquires information. It can effectively characterize the low-frequency region of speech, but it cannot be completely accurate in the presence of background noise; also, the recognition accuracy in mid-to-high frequencies is not high. Since there are more Mel filters in the low-frequency region and fewer in the high-frequency region, it can better characterize low-frequency speech signals. However, a driver’s negative emotion signal is often a high-frequency signal; hence, it is necessary to design the Mel filter bank to better characterize the high-frequency signal. Therefore, the Mel filter bank can be reversed to obtain the inverse Mel filter bank, and the IMel-frequency cepstrum coefficient (IMFCC) is used to make it denser in the high-frequency region. The frequency power analysis of negative emotional signals is set at the highest frequency of the filter to realize the characterization of high-frequency features.

Among them, the relationship between $f_{1-Mel}$ of the IMFCC and sound frequency f can be approximated with Equation (6).
(6)$$f_{1-Mel}=2840-1125\times \ln\left(1+\frac{8000-f}{700}\right)$$

Compared with the MFCC, the filter frequency response of the IMFCC is shown in Equation (7).
(7)$$H_{i}(k)=\;H_{M+1-i}(\frac{N}{2}+1-k),1\leq i\leq M$$

In Equation (7), $H_{i}(k)$ is the frequency response of the i-th inverse Mel filter compared to the M + 1-i-th Mel filter, and M is the number of filters.

Therefore, for the feature extraction of audio modal data, the combination of the MFCC and IMFCC groups is used to extract features of different frequency speech signals, and the hybrid MFCC is obtained to improve the performance of classification or recognition tasks. The hybrid MFCC feature vector combines a variety of MFCC features with different frequencies, which can improve the accuracy and stability of the task. In this paper, the 1–6 order coefficients of the MFCC and the 7–12 order coefficients of the IMFCC are spliced to obtain a 12-order spliced MFCC. Figure 10 shows the Mel frequency–frequency corresponding curve of the spliced MFCC characteristic parameters. Figure 11 shows the hybrid MFCC feature extraction process.

#### 4.2.2. HOG Extracts CK+ Dataset Features

The HOG algorithm uses the image normalization method to simplify the calculation of intensity; then, for the image sample input into the algorithm, it will accurately calculate the gradient features of the local space of the sample, and these features include the specific distribution of intensity and direction. This important idea has achieved great success in the field of pedestrian detection. This paper applies this idea to the field of emotion detection as a shallow feature extractor for image samples.

The HOG feature extraction is used for CK+ data. The HOG algorithm divides the image into several small cells, calculates the histogram of the gradient direction of pixels in each cell, and connects these direction histograms to obtain a global feature vector. Represents the texture and shape information of an image. In the facial expression recognition task, these features can be used to describe the local texture and contour information of facial expressions, to realize the recognition and classification of facial expressions. The HOG feature extraction process includes the steps shown in Figure 12.

In the step of outputting the HOG feature vector, all blocks in the detection window will be traversed to obtain the complete gradient direction histogram of the detection window, which is the HOG feature vector we need. In order to experience the HOG features more intuitively, we use the code to visualize the HOG features, and the results are shown in Figure 13 below.

After the above steps of image preprocessing, we converted the video sequences in the CK+ dataset into experimental samples suitable for subsequent neural networks, and the data format was also converted from video sequences into multidimensional arrays that are convenient for computer calculations.

#### 4.2.3. Circular LBP Algorithm to Extract Features of NTHU Dataset

The LBP is an operator used to describe local features of an image. The LBP features have significant advantages such as gray-scale invariance and rotation invariance. It was proposed by T. Ojala, M. Pietikäinen, and D. Harwood in 1994 [69]. As the LBP feature calculation is simple and the effect is good, the LBP feature has been widely used in many fields of computer vision, and the LBP feature is relatively famous. The application is used in face recognition and object detection.

The original LBP operator is defined in the area of 3 × 3 pixels, with the center pixel of the area as the threshold, the gray value of the adjacent 8 pixels is compared with the pixel value of the area center, if the surrounding pixels are greater than the center pixel value, then the position of the pixel is marked as 1; otherwise, it is 0. In this way, the 8 points in the 3 × 3 area can be compared to generate an 8-bit binary number, and these 8-bit binary numbers are arranged in turn to form a binary number. This binary number is the LBP value of the central pixel, and the LBP value of the central pixel reflects the pixel. Figure 14 shows the texture information of the image processed in the traditional LBP mode.

Equation (8) is a more formal definition.
(8)$$LBP(x_{c},y_{c})=\sum_{p=0}^{P-1}2^{P}S(i_{p}-i_{c})$$

Among them, $\left(x_{c},y_{c}\right)$ represents the central element of the 3 × 3 area and its pixel value is $i_{p}$; $i_{c}$ represents the values of other pixels in the neighborhood. The s(x) is a symbolic function, defined as Equation (9):(9)$$s(x)=\left\{\begin{array}{c} 1\;\;,\;if\;\;x\geq 0\\ 0,\;\;\;\;\;\;\;\;\;\;\;\;else \end{array}\right.$$

The above-mentioned LBP operator only includes the area within a small fixed nine-square grid for the driver’s facial image.

Such complex image data cannot meet the needs of multiple scales. Therefore, based on the limitations of the traditional LBP algorithm, Ojala et al. proposed the circular LBP algorithm. The traditional LBP mode is to extract the LBP feature value through the 3 × 3 area, which cannot realize the extraction of image feature textures at different scales, while the circular LBP algorithm can cover texture extraction at different scales. As shown in Figure 15, the 3 × 3 area of the traditional LBP is changed into a circular area, the radius of which is R, the number of sampling points is T, and the sampling method is equidistant sampling. R and T can be changed. When R is 1 and T is 8, when R is 2 and T is 8, and when R is 2 and T is 16, and the LBP eigenvalue is expressed as ${LBP}_{R,T}$. For the NTHU dataset, this paper uses the circular LBP feature extraction method to concentrate face images for feature extraction.

Obtaining the circular LBP eigenvalue of the center point is similar to the acquisition of the traditional LBP eigenvalue, as shown in Formulas (10) and (11) that follow.
(10)$$S(g_{T}-g_{m})=\left\{\begin{array}{c} {1\;,\;g}_{T}\geq g_{m}\\ {1\;,\;g}_{T}<g_{m} \end{array}\right.$$
(11)$$LBP=\sum_{1}^{T}S(g_{T}-g_{m})2^{P}$$

Among them, T is the number of sampling points, $g_{m}$ is the gray value of the center point, $g_{T}$ is the gray value of each sampling point, and each circular LBP feature is counted to form a circular LBP histogram through the circular LBP algorithm histogram for identification. Figure 16 shows the circular LBP feature image when R is 2 and T is 8.

### 4.3. Evaluation Index

According to the discrete emotion model, there are six driver emotion classification tasks: sadness, anger, fear, fatigue, happiness, and neutrality. The emotion classification task uses accuracy and the F1 Score as the evaluation indicators of model performance. For binary classification problems, i.e., classifying instances as positive or negative, four situations arise. If the situation is a positive example and is also predicted to be a positive example, it is a true-positive (TP); if the situation is a negative example and is predicted to be a positive example, it is called a false-positive (FP). Correspondingly, if the situation is a negative example and is predicted to be a negative example, it is called a true-negative (TN) if the positive class is predicted to be a negative example, then it is a false-negative (FN). Obviously, TP + TN + FP + FN = total number of samples. The “confusion matrix” of the classification results is shown in Table 3.

Accuracy is the most common evaluation index, which is the percentage of correct prediction results in the total samples, as shown in Formula (12) below.
(12)$$Accuracy=\frac{TP+TN}{TP+TN+FP+FN}$$

Although the accuracy rate can judge the overall correct rate, it cannot be used as a good indicator to measure the result when the sample is unbalanced. That is, if the samples are not balanced, the accuracy rate will be invalid. Therefore, this article also refers to two other indicators: precision and recall. Precision is also called the precision rate. It refers to the prediction results. It is defined as the probability of the actual positive samples among all the samples that are predicted to be positive. It calculates how many confidence predictions that are in the results of the predicted positive samples are correct, as shown in Equation (13) below.
(13)$$Precision=\frac{TP}{TP+FP}$$

Recall is also called the recall rate, it is used for the original sample, and it is defined as the probability of a situation predicted as a positive sample in the actual positive sample, as shown in Formula (14) below.
(14)$$Recall=\frac{TP}{TP+FN}$$

Through the above formula, it can be observed that the numerators is the same as recall, with both of them being TP (true positive); however, the denominators are different. One denominator is TP + FP (false positive), and the other denominator is TP + FN (false negative). The recall and F1 score are conflicting measures. Generally, when the F1 score is high, recall tends to be low, and vice versa. Therefore, in order to balance the importance of recall, this paper simultaneously considers both recall and precision to find a balanced value, which is known as the F1 score. Its definition is given by Formula (15).
(15)$$F1=2\cdot \frac{Precision\cdot Recall}{Precision+Recall}$$

### 4.4. Experimental Results

This paper constructed a CNN + Bi-LSTM + HAM network. In the experiment, we used the Adam algorithm to optimize the gradient descent after each network layer. The Adam algorithm [70] can accelerate the convergence speed of the model, effectively reduce the training time and improve the training efficiency. We divided the dataset into a training set and a test set according to a certain ratio; then, we performed training on the training set, and verified the performance of the trained model in the test set. The RAVDESS and CK+ datasets were divided into the training set according to a ratio of 8:2 and the test sets were combined with the NTHU’s training set and evaluation verification set to form the training set and verification set of this paper. Through repeated parameter tuning and other operations, we conducted 100 rounds of training. At the same time, in the training process, in order to limit the overfitting in the training phase and improve the generalization ability of the model, this paper adopted the cross-validation method to assist in adjusting the model parameters, aiming to obtain a more stable and better-performing model.

The hardware configuration of the model training in this paper was the NVIDIA Tesla V100 GPU with 32 GB video memory, the operating system was Ubuntu20.04, and the deep learning framework was Py Torch1.8. During the training process, the Adam optimizer was used, the batch size was 64, and the learning rate decay strategy was cosine decay. When training the dataset, the initial learning rate was set to 5 × 10^−2^, the learning rate hot restart epoch was 5, and the minimum learning rate was 1 × 10^−8^. The total training epoch was 100.

After 100 rounds of training, the comparison of loss function values and accuracy values on the training set and validation set is shown in Table 4.

From the results in the above table, we can see that after 100 rounds of training, the loss function value of the model on the training set dropped from 1.8361 to 0.5306, and the accuracy increased from 0.1695 to 0.8203. The loss function value on the validation set also dropped from 1.9524 to 0.4574, and the accuracy increased from 0.1452 to 0.8552. This shows that the generalization ability of the model was successfully improved while maintaining an appropriate model complexity. In order to better illustrate the changing trend of these two datasets during the training process, Figure 17 shows the loss function value and the accuracy value curves of the training and verification sets of the 1–100th round.

Confusion matrix: in order to better demonstrate the recognition of each expression using the CNN + Bi-LSTM + HAM network model proposed in this paper, Table 5 shows the confusion matrix of the drivers’ six-category emotion recognition results when the recognition performance is optimum.

As shown in Figure 18, we obtained the confusion matrix diagram of drivers’ comprehensive emotion recognition, which was used for the specific classification of the following six emotions: sadness, anger, fear, fatigue, happiness, and neutral. As can be seen from the confusion matrix, the recognition model had the highest recognition accuracy of 88% for frustrated and fearful expressions. The recognition accuracy of anger, happiness, and neutral emotions were 85%, 86%, and 87%, respectively. Fatigue’s accuracy in identifying emotions was 79%. The reason is that the NTHU dataset as the fatigue label module was not obtained under strict laboratory conditions. At the same time, the two subsets selected in this article and the samples in the CK+ dataset had uneven data distributions.

Among them, the recognition of six types of emotions is the data of the diagonal line shown in the table, and the overall experiment reached 0.8552. In order to verify the effectiveness of the proposed scheme in this paper, this paper did a comparative experiment under the same dataset processing, in which the CNN + LSTM network reached 0.7980, the CNN + Bi-LSTM network reached 0.8058, the CNN + Bi-LSTM + CAM network reached 0.8227, and the CNN + Bi-LSTM + SAM network reached 0.8275. At the same time, in order to balance the two key evaluation indicators and make them reach the highest at the same time, the F1 score was used for model evaluation. The comparison experiments of each model and the comparison of the F1 score results are shown in Table 6.

Use the structure diagram of the discrete emotion experiment to better display the comparative test results. The comparison results of each model are shown in Figure 19a, and the F1 score results are shown in Figure 19b. The higher the index and the F1 score, the better. It can be seen that the performance of our proposed method (CNN + Bi-LSTM + HAM) was stronger than that of the benchmark method, which fully demonstrates the effectiveness of the proposed scheme for drivers’ six-category emotion recognition.

In addition, after the model training was completed, in order to verify the reliability of the framework proposed in this article, we also added a discussion on the results of the current excellent single-modal emotion datasets in emotion recognition; at the same, we time made up for the imbalanced distribution of the datasets used in this article. The limitations were verified on the public test set of the FER2013 dataset. Due to the six categories of drivers’ comprehensive emotions determined in this article, we selected the five emotions of sadness, anger, fear, happiness, and neutral from the public test set of the FER2013 dataset. Combined with the comparative experiments on the FER2013 dataset, we proved that the proposition in this article demonstrated the effectiveness of the model. Among them, in order to improve the transfer generalization effect on the FER2013 dataset, we added a fine-tune process for the FER2013 dataset based on the model. Figure 20 shows the confusion matrix of the five-category emotion recognition results of our model on the public test set of the FER2013 dataset.

The network proposed in this article was verified using the public test set of the FER2013 dataset. The five-category emotion recognition effect determined on the FER2013 dataset was effectively improved compared to the VGGNET network that performed more advanced tasks in the FER2013 dataset. Compared to the accuracy of the traditional CNN + SVM and VGG + SVM networks, the models in this paper were greatly improved [71,72]. Table 7 compares the experimental results of the scheme proposed in this article with the above-used comparison scheme.

After comparison, the accuracy of the CNN + Bi-LSTM + HAM on the FER2013 dataset was higher than the benchmark model solution, which verifies the generalization of the framework proposed in this article and is also reliable for emotion recognition of single-modal emotion datasets; furthermore, it considers the limitations of the imbalanced distribution of the dataset used in this article from the side. The experimental results of the model showed that the CNN + Bi-LSTM + HAM driver emotion recognition framework can obtain good detection results on the dataset selected in this article. By using a framework composed of the CNN, Bi-LSTM, and HAM, the model can extract meaningful features from the input data and achieve high accuracy and performance in driver emotion recognition tasks.

## 5. Conclusions

In this paper, we utilize audio, image, and video modality data from three different datasets, the RAVDESS, CK+ and NTHU, and build a driver emotion recognition model by extracting and fusing their features. We use algorithms such as the hybrid MFCC, HOG, and circular LBP to extract features from audio, image, and video data, respectively. Then, these heterogeneous features are spliced together, input into the CNN + Bi-LSTM + HAM model; afterwards, the hybrid attention module is used to focus on key features to achieve effective classification and accurate identification of the driver’s emotional state. The results show that, compared with other models, the method in this paper can achieve higher six-category accuracy, which verifies the reliability of the network proposed in this paper.

Due to the constraints of research time and measurement methods, there are still some limitations in this paper, which need to be further improved in future research works, mainly in the aspects that follow.

In order to make the experimental data more real and effective, the training set, verification set, and testing in the experiment were all completed using existing labeled experimental data and were not applied in the real environment. At the same time, different categories of samples in different datasets caused an uneven distribution. In later research, we will consider collecting more data from real scenarios and study the application of other attention mechanisms, in hope of further improving the generalization ability of the model.

The dataset selected in this article will have an adverse impact on model performance due to an uneven category distribution. To address this problem, in the dataset preprocessing, we calculated the sample size distribution of each category and found the data imbalance phenomenon; during model training, we adopted an oversampling strategy to oversample categories with few samples to mitigate data imbalance. However, we also realize that there are certain problems with oversampling. In the future, we will specialize in data augmentation, generative methods, etc. to further alleviate this problem.

The RAVDESS dataset used in this article was originally designed to evaluate speech expressions of emotion, and is not specifically designed for driving scenarios. This paper chose this dataset because it contains rich speech samples and annotated emotion labels. Although this dataset is not entirely designed for driving situations, the types of emotions it covers are highly correlated with emotional states related to safe driving. This provides an effective data source for training to identify driver emotional states. The possible impact of the dataset not being fully designed for the driving environment should be considered in subsequent research works.

This article uses different datasets, each of which is independent. In the actual driving environment, the information flow corresponding to the multi-modality may conflict with the recognition results. This article has not yet considered the conflicts that may occur when integrating multi-modal features. This is an obvious flaw. In the follow-up research work, we will consider designing a model output adjustment module to automatically adjust its weights to balance the results when the results of different modal features deviate significantly. We will also continue to study multi-source heterogeneous features. An integration method was used to handle the conflicts between various modes and improve the generalization ability of the model in complex actual environments. At the same time, this study verified the effectiveness of the method on a single dataset, laying a foundation for future research. The next work will focus on the integration and reconciliation of multi-source features to promote the feasibility of this method in practical applications.

The experimental results of this article recognize the six types of driver emotions, which are not sufficient for the complexity of the environment and driving situations. A smaller and more efficient model should be considered for driver emotion recognition. Furthermore, in the future, for a monitoring and early warning reminder system of drivers’ emotional state, an effective deep learning model should be added to carry out specific grading identification of drivers’ emotions. Through the emotion grading results, corresponding driving adjustment strategies should be adopted to improve driving safety.

## Figures and Tables

**Figure 1 sensors-23-08293-f001:**
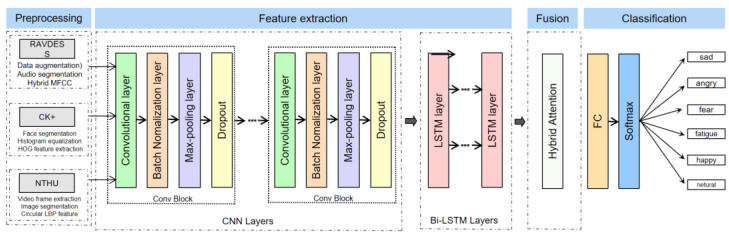
CNN + Bi-LSTM + HAM network driver comprehensive emotion recognition framework.

**Figure 2 sensors-23-08293-f002:**
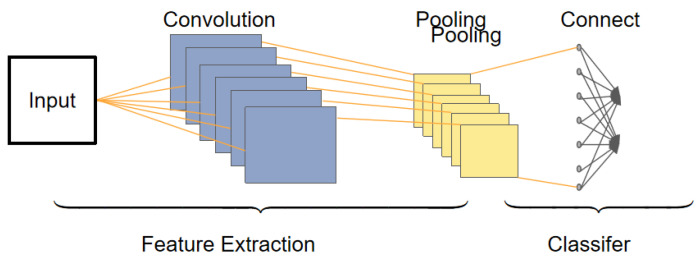
CNN network structure.

**Figure 3 sensors-23-08293-f003:**
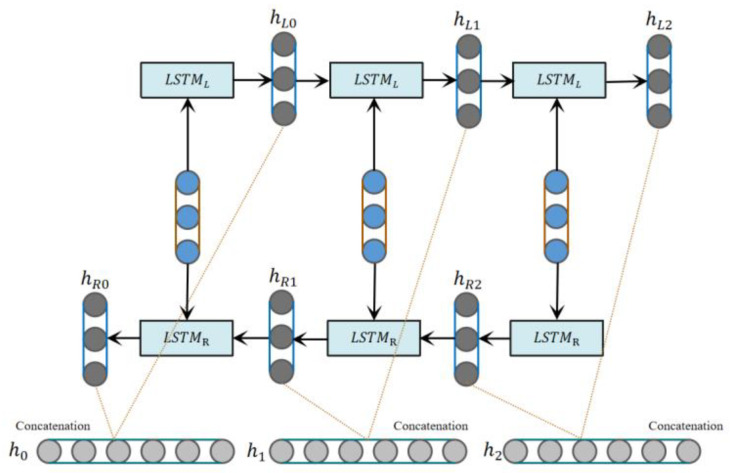
Bi-LSTM network structure diagram.

**Figure 4 sensors-23-08293-f004:**
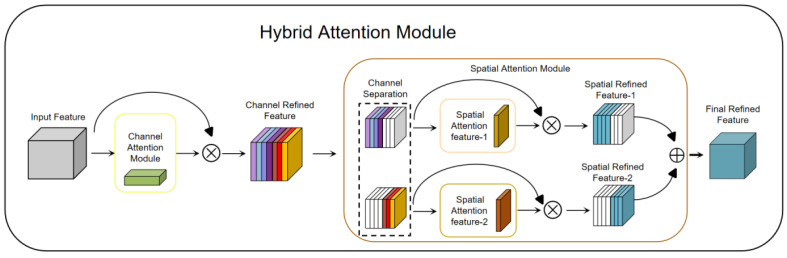
Hybrid attention module network structure diagram.

**Figure 5 sensors-23-08293-f005:**
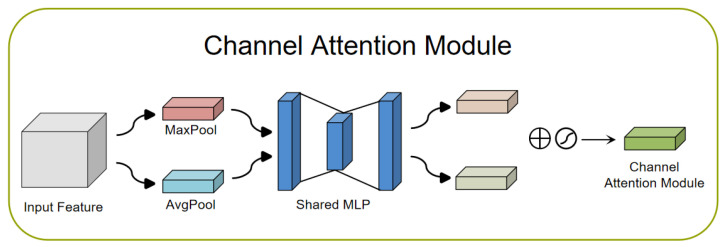
Channel attention module network structure diagram.

**Figure 6 sensors-23-08293-f006:**
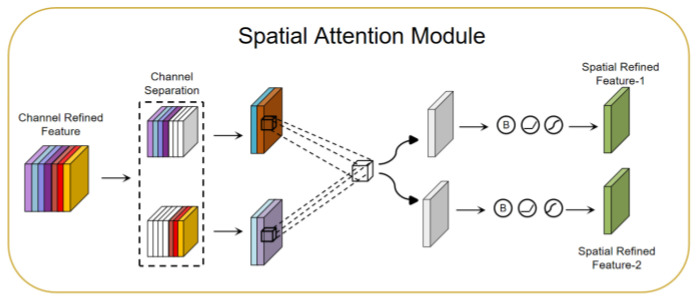
Spatial attention module network structure diagram.

**Figure 7 sensors-23-08293-f007:**
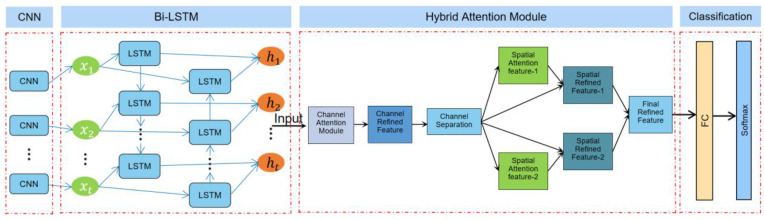
CNN + Bi-LSTM + HAM network model structure diagram.

**Figure 8 sensors-23-08293-f008:**
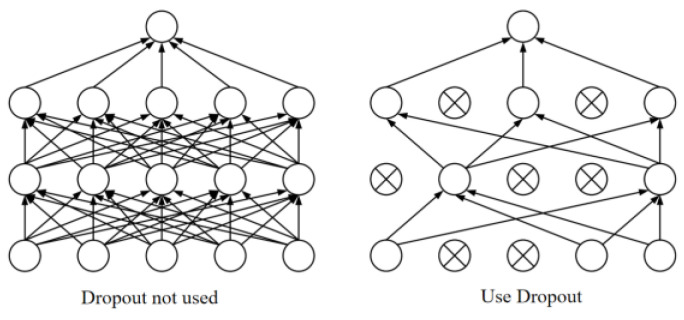
Dropout working diagram.

**Figure 9 sensors-23-08293-f009:**
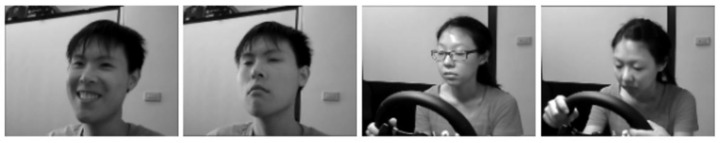
Simulated driving conditions of different people in different scenarios.

**Figure 10 sensors-23-08293-f010:**
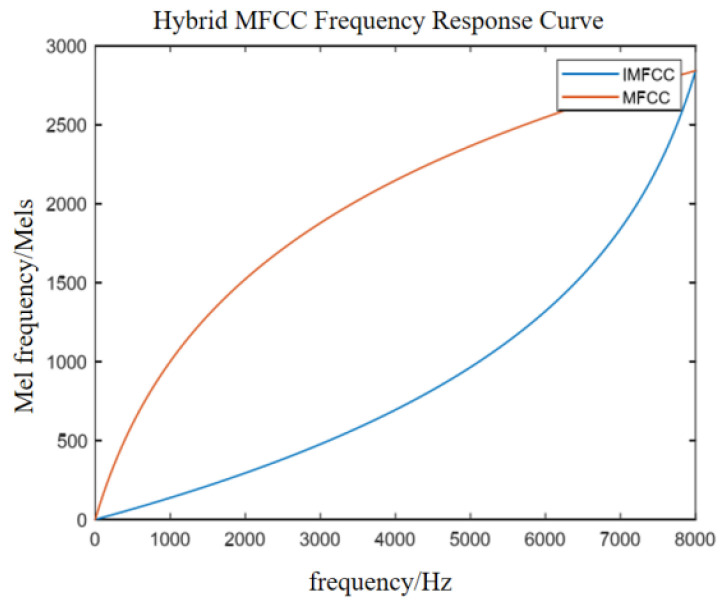
Hybrid MFCC frequency conversion relationship.

**Figure 11 sensors-23-08293-f011:**
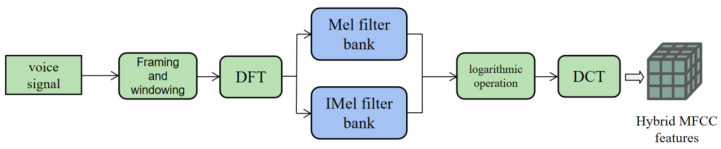
Hybrid MFCC feature extraction process.

**Figure 12 sensors-23-08293-f012:**
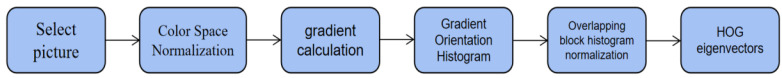
HOG feature extraction flow chart.

**Figure 13 sensors-23-08293-f013:**
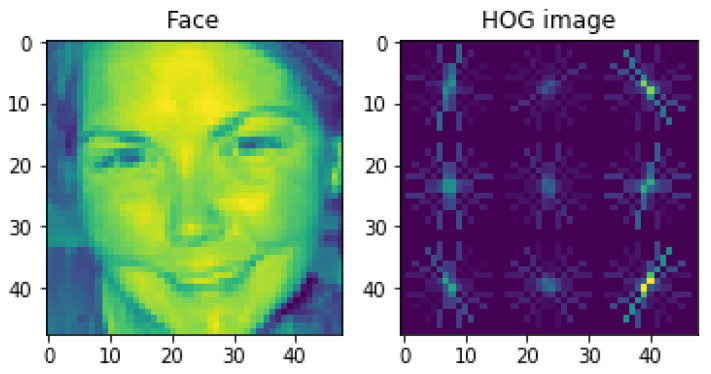
HOG feature visualization.

**Figure 14 sensors-23-08293-f014:**
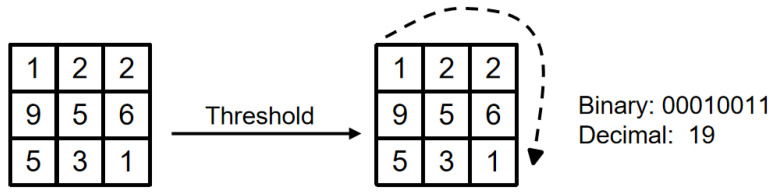
Traditional LBP operator.

**Figure 15 sensors-23-08293-f015:**
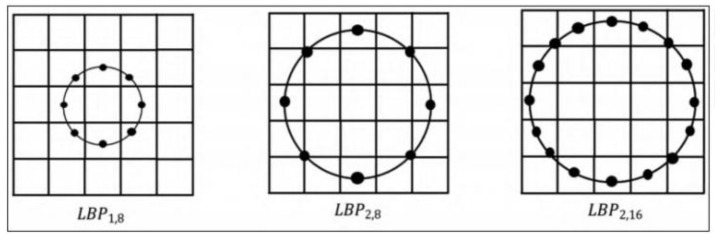
Circular LBP feature extraction.

**Figure 16 sensors-23-08293-f016:**
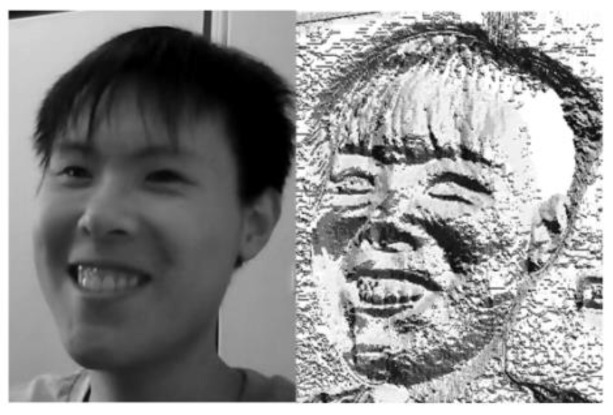
Circular LBP feature image when R is 2 and T is 8.

**Figure 17 sensors-23-08293-f017:**
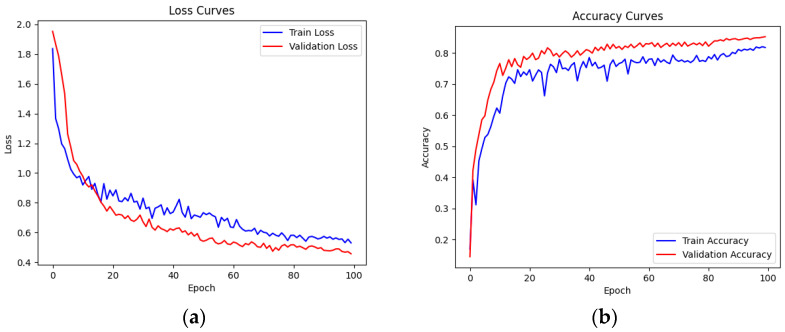
Loss function curves and accuracy curves on the training set and validation set. Picture (**a**) shows the loss function curve on the training set and verification set, and picture (**b**) shows the accuracy curve on the training set and verification set.

**Figure 18 sensors-23-08293-f018:**
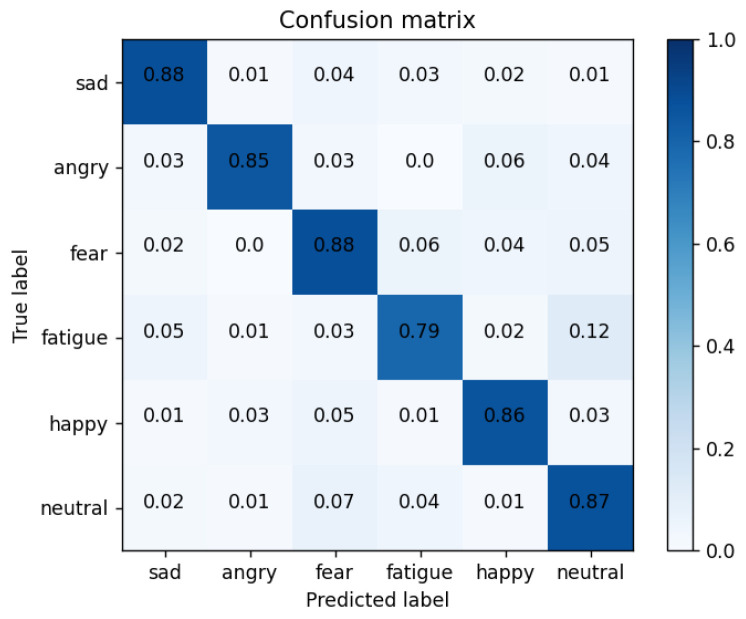
Confusion matrix of our method.

**Figure 19 sensors-23-08293-f019:**
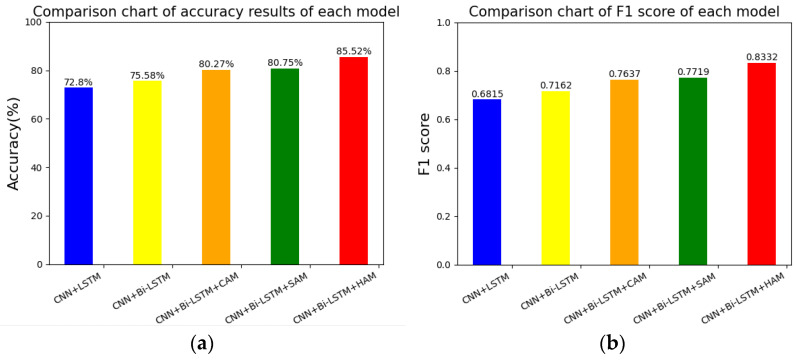
Accuracy and F1 score of each model. Picture (**a**) shows the accuracy of each model, and picture (**b**) shows the F1 score of each model.

**Figure 20 sensors-23-08293-f020:**
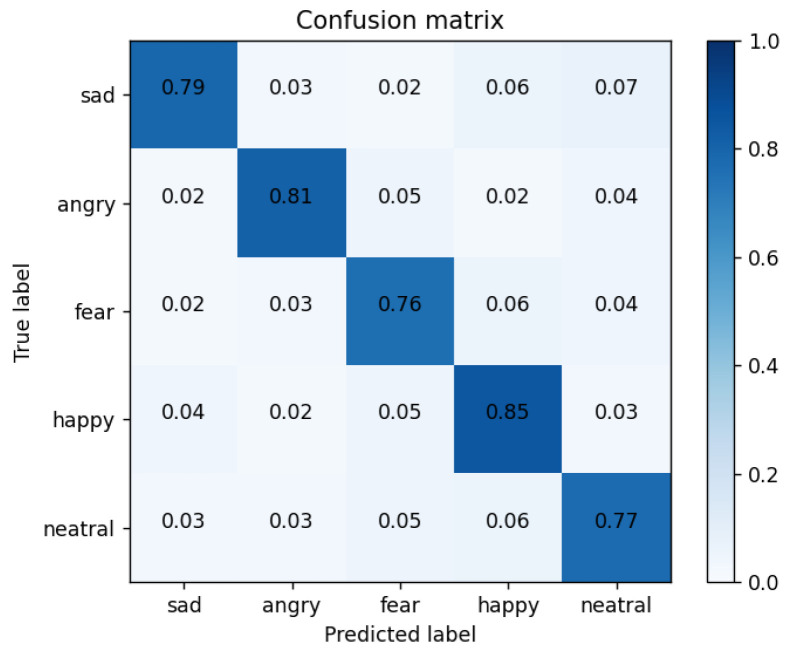
Five-category emotion confusion matrix of our method on the FER2013 dataset.

**Table 1 sensors-23-08293-t001:** RAVDESS dataset sentiment label distribution table.

Emotional Label	Happiness	Sadness	Anger	Fear	Disgust	Surprise	Calm	Neutral	Total
Male	96	96	96	96	96	96	96	48	720
Female	96	96	96	96	96	96	96	48	720
Total	192	192	192	192	192	192	192	96	1440

**Table 2 sensors-23-08293-t002:** Distribution table of seven emotional labels in CK+ dataset.

Emotional Label	Happiness	Sadness	Anger	Fear	Disgust	Surprise	Neutral
The number of occurrences	69	28	46	25	59	83	146

**Table 3 sensors-23-08293-t003:** Confusion matrix of classification results.

Forecast Result		
The True Situation	Positive	Negative
Positive	TP	FN
Negative	FP	TN

**Table 4 sensors-23-08293-t004:** Comparison table of loss function values and accuracy values on the training set and verification set.

Number of Training Rounds	Training Set Loss	Training Set Accuracy	Validation Loss	Validation Set Accuracy
Round 1	1.8361	0.1695	1.9524	0.1452
…	…	…	…	…
Round 100	0.5306	0.8203	0.4574	0.8552

**Table 5 sensors-23-08293-t005:** Confusion matrix of accuracy rate of drivers’ six-category emotion recognition.

	Sadness	Anger	Fear	Fatigue	Happiness	Neutral
sad	0.88	0.01	0.04	0.03	0.02	0.01
angry	0.03	0.85	0.03	0	0.06	0.04
fear	0.02	0	0.88	0.06	0.04	0.05
fatigue	0.05	0.01	0.03	0.79	0.02	0.12
happy	0.01	0.03	0.05	0.01	0.86	0.03
neutral	0.02	0.01	0.07	0.04	0.01	0.87

**Table 6 sensors-23-08293-t006:** Comparison of accuracy and F1 score results of each model.

Model	Precision	F1 Score
CNN + LSTM	72.80%	0.6815
CNN + Bi-LSTM	75.58%	0.7162
CNN + Bi-LSTM + CAM	80.27%	0.7637
CNN + Bi-LSTM + SAM	80.75%	0.7719
CNN + Bi-LSTM + HAM	85.52%	0.8332

**Table 7 sensors-23-08293-t007:** Model performance comparison table of CNN + Bi-LSTM + HAM and other solutions on the FER2013 dataset.

Model Scheme	Accuracy
CNN + SVM	68.57%
VGG + SVM	71.18%
VGGNET	76.39%
CNN + Bi-LSTM + HAM	79.60%

## Data Availability

This study investigated the publicly available RAVDESS, CK+, NTHU, and FER2013 datasets.

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
