# Peer review of "Drivers’ Comprehensive Emotion Recognition Based on HAM"

_sensors, 2023, doi:10.3390/s23198293_

Round 1

Reviewer 1 Report

45-46: Please provide multiple references when you quote that something has "received more and more attention"

46 - 48, 50-54: Please cite the studies here

82-84: please cite those previous studies with limitations that you mention

316: Define / Abbreviate SAM with citations if possible

429: "Ted"?

788: accuracy increased. Missing "accuracy"

Table 4: Could you please provide a regular confusion matrix as it provides more info than the matrix provided here?

The article works with different datasets, where each of them are independent. However, as the article intends to introduce this method for real world where these multi-modal data is simultaneously available, it would be better if the article can highlight potential issues and propose solutions for such cases. For instance, what would happen if there were to be conflicting results from each of these information streams for the same event, etc?

Please quote and cite the results of other similar studies that were performed on similar datasets, to showcase the performance improvement that can be achieved with the proposed model.

Please double check for grammatical errors and proper sentence construction. For instance, lines 10-12, provide vs provides changes the meaning of the sentence. Please correct it to provides or rephrase the sentence.

Author Response

Dear reviewer, thank you for your patient guidance and suggestions. Please find our reply in the attachment.

Reviewer 2 Report

In this work the authors present a multi-modal system that performs emotion recognition in driving context.

The topic appears to be interesting, but the manuscript still has aspects to be improved. Below the list of the main concerns I have :

The literature part needs to be more detailed. Information about used databases need to be included.

The choice of using RAVDESS for speech emotion recognition need to be justified. As it is not designed for drivers emotions states, the database may not be adequate for the context. same for other databases. 

Authors should employ databases that are used in literature and should include the performance comparison with that in literature for single modalities to underline the importance of the proposed approach. 

When using abbreviation, the full name should be mentioned at earliest.

The contribution in the driving context is not coming out clearly as the used databases that are used are not related to the field. Also, there is a lack of comparison with state of the art works

Author Response

(The authors gave the same response as above.)

Reviewer 3 Report

In the article, the authors address the challenging task of driver emotion recognition through multimodal data fusion. While the article has several strengths, it also has notable limitations that warrant consideration.

Strengths:

1) The article adopts a holistic method by combining audio, image and video data for driver emotion recognition. This multimodal strategy fits well with real-world scenarios where emotions manifest through different channels. The fusion of these modalities increases the robustness of the model.

2) The article introduces and applies effective feature extraction techniques, such as hybrid MFCC, HOG and circular LBP, tailored to each data modality. This demonstrates a comprehensive understanding of feature engineering, which is crucial for improving classification performance.

3) A notable innovation is the integration of the Hybrid Attention Module. HAM enhances the model's focus on essential features, contributing significantly to improved classification accuracy. This innovative feature is a valuable addition to the proposed model.

4) The authors carry out a rigorous experimental evaluation involving 100 training rounds, using appropriate metrics such as accuracy, precision, recall and F1 score. This extensive evaluation provides substantial evidence for the effectiveness of the model.

5) The article provides a comparative analysis with other models, showing that the proposed CNN+Bi-LSTM+HAM model outperforms benchmark methods. This comparative evaluation adds credibility to the proposed article.

Weaknesses:

1) Data imbalance: A major limitation of the study is the imbalance in the distribution of the datasets, which is particularly evident in the NTHU dataset. Data imbalance can adversely affect the performance of the model, as some classes have more samples than others. Future research should focus on addressing this issue, possibly through data augmentation or resampling techniques.

2) Limited discussion of single modality results: The article lacks a discussion of current and best single modality results in emotion recognition. For example, in the context of the video modality, the article could benefit from including a discussion of the top (first) three results for each dataset (for 7 and 8 basic emotions). In particular, AffectNet is a widely recognised corpus for emotion recognition in the video modality, and its state of the art (SOTA) results should be referenced (SOTA for AffectNet [7 emotions]: https://paperswithcode.com/sota/facial-expression-recognition-on-affectnet, including POSTER++ (2023), Emotion-GCN (2021), EmoAffectNet (2022)). This additional context would enrich the references section and provide a broader perspective by referencing previous work in the global research community (2020-23).

Conclusion:

In conclusion, the article presents a promising approach to driver emotion recognition through multimodal data fusion. The article shows strengths in its multimodal approach, feature extraction methods and the innovative Hybrid Attention Module. However, it also faces challenges related to data imbalance and the lack of discussion of state-of-the-art results in single modality analysis. Despite these limitations, the article's rigorous experimental evaluation and comparative analysis highlight the effectiveness of the proposed model. With further refinement and inclusion of references to the best results in single modality emotion recognition, this approach has the potential to make a significant contribution to the field of driver emotion recognition.

Recommendation: The authors are encouraged to address the identified weaknesses by considering strategies to mitigate data imbalance and incorporating discussions of state-of-the-art results in single modality emotion recognition. These improvements would enhance the applicability of the model and contextualise it within the broader research landscape.

Moderate editing of English language required.

Author Response

(The authors gave the same response as above.)

Round 2

Reviewer 2 Report

Abbreviations need to be defined first : ECG, EEG, EMG, MFCC, LBP, MSE, DFT, IMFCC

The sentence is not clear in page 2 : ...and proposed a good research method in enhancing emotion recognition, effectively 53 improving driving safety sex[11-13].

Author Response

Dear reviewers, please see the attachment for our response.

Reviewer 3 Report

Overall, the authors have made improvements to the article and addressed various questions. However, I'm unsure about the criteria they used when including references 9 and 10 (ranked 1st and 4th on paperwithcode) in their article. Interestingly, they overlooked references ranked 2nd and 3rd in the evaluation of the 7 basic emotions on the AffectNet corpus.

Upon closer inspection, it becomes apparent that reference number 10 in the citation list actually holds the 2nd position when considering 8 basic emotions. This discrepancy arises because the rankings do not include the Emotion-GCN and EmoAffectNet methods, both of which outperform Multi-task EfficientNet-B2 in the evaluation of the 7 basic emotions. For the sake of completeness, it would be advisable for the authors to also incorporate information about Emotion-GCN and EmoAffectNet into their article, given their superior performance compared to Multi-task EfficientNet-B2.

Minor editing of English language required.

Author Response

(The authors gave the same response as above.)
